# HIV-1 infection enhances innate function and *TLR7* expression in female plasmacytoid dendritic cells

Flora Abbas[1], Claire Cenac[1], Ali Youness[1], Pascal Azar[1], Pierre Delobel[1,2], Jean-Charles Guéry[1]

**Plasmacytoid dendritic cells (pDCs) express TLR7, a ssRNA-sensor encoded on the X chromosome, which escapes X chromosome inactivation (XCI) in females. pDCs are specialized in the production of type 1 interferons (IFN-I) through TLR7 activation which mediates both immune cell activation and also reactivation of latent HIV-1. The effect of HIV-1 infection in women under antiretroviral therapy (ART) on pDC functional responses remains poorly understood. Here, we show that pDCs from HIV/ART women exhibit exacerbated production of IFN-α and TNF-α compared with uninfected controls (UC) upon TLR7 activation. Because *TLR7* can escape XCI in female pDCs, we measured the contribution of *TLR7* allelic expression using SNP haplotypic markers to rigorously tag the allele of origin of *TLR7* gene at single-cell resolution. Herein, we provide evidence that the enhanced functional response of pDCs in HIV/ART women is associated with higher transcriptional activity of the *TLR7* locus from both X chromosomes, rather than differences in the frequency of *TLR7* biallelic cells. These data reinforce the interest in targeting the HIV-1 reservoir using TLR7 agonists in women.**

## Introduction

Despite the efficacy of long-term antiretroviral therapy (ART), HIV-1 latent reservoir persists in individual living with HIV-1 and can reactivate and cause active infection upon treatment interruption (Mitchell et al, 2020). HIV persists in a small pool of cells, particularly memory CD4 T cells, harboring integrated and replication-competent viral genomes (Dufour et al, 2020). Latency removal agents have been developed as potential means to reduce the HIV-1 reservoir by inducing reactivation of latent proviruses followed by elimination of the resulting infected cells by the immune system (Dufour et al, 2020). Second generations of latency removal agents have been developed that stimulate the endosomal Toll-like receptors TLR7, TLR8, or TLR9, which are of high interest as potential cure strategies because of their capacity to both reverse HIV-1

latency and enhance HIV-1-specific immune control (Borducchi et al, 2016, 2018; Lim et al, 2018; Macedo et al, 2019; Meas et al, 2020). Transient reactivation of the latent simian immunodeficiency virus reservoir and sustained viral remission have been obtained using the TLR7 agonist ligand GS-9620 in simian immunodeficiency virus–infected rhesus macaques under ART alone (Lim et al, 2018) or in combination with anti-envelope antibodies or therapeutic vaccines (Borducchi et al, 2016, 2018).

Plasmacytoid dendritic cells (pDCs) express TLR7, which senses ssRNA, and induce the secretion of copious amounts of type I IFNs (IFN-I) that promote cell-autonomous antiviral defense through interferon inducible genes and also serves as bridge to potentiate innate and adaptive immunity promoting antibody responses and enhancing cytotoxic T-cell and NK cell potential (Gonzalez-Navajas et al, 2012). TLR7 agonist ligands can mediate both immune cell activation and HIV-1 expression in cells from HIV-1–infected individuals on suppressive ART (HIV/ART) (Tsai et al, 2017; Van der Sluis et al, 2020). In this model, the reactivation of the latent HIV-1 reservoir in PBMCs from ART-suppressed individuals was dependent on pDC-mediated production of IFN-α which was strongly induced in HIV-infected PBMCs (Tsai et al, 2017). Understanding the mechanisms controlling the expression of TLR7 and the production of IFN-I is therefore an important issue in HIV-1 infection, in particular for the development of new strategies for viral eradication (Borducchi et al, 2016, 2018; Lim et al, 2018; Macedo et al, 2019). Although pDC numbers are reduced in the blood compartment of HIV/ART (Kamga et al, 2005; Fontaine et al, 2009; Sabado et al, 2010), limited studies have been performed to assess their functional properties in response to TLR7 agonist ligands, particularly in women.

Herein, we ex vivo assessed the innate function of pDCs upon TLR7 stimulation by controlling for the presence of the rs179008 single-nucleotide polymorphism c.32T allele which inhibits TLR7 protein expression and pDC innate functions, selectively in female, but not in male (Azar et al, 2020; Guéry, 2021). We show here that HIV/ART women exhibited exacerbated production of IFN-α and TNF-α as compared with uninfected controls (UC). Because *TLR7* can escape X chromosome inactivation (XCI) in human female pDCs (Souyris et al, 2018), we then assessed the contribution of *TLR7*

---

[1]Institut Toulousain des Maladies Infectieuses et Inflammatoires (INFINITY), Université de Toulouse, Institut National de la Santé et de la Recherche Médicale, Centre National de la Recherche Scientifique, Université Paul Sabatier, Toulouse, France   [2]Service des Maladies Infectieuses et Tropicales, CHU Purpan, Toulouse, France

Correspondence: jean-charles.guery@inserm.fr

 

allelic expression on the functional reprogramming of pDCs in HIV/ART women, using SNP haplotypic markers to rigorously tag the allele of origin of *TLR7* gene at single-cell resolution.

# Results

### The *TLR7* rs179008 c.32T allele inhibits IFN-α production by pDCs from HIV-1–infected women under ART

We recently established that the *TLR7* rs179008 c.32T allele, which the frequency distribution was similar between HIV-1–infected and control women (Azar et al, 2020), is a sex-specific protein expression quantitative trait locus (pQTL) where c.32T allele carriage was associated with both impaired TLR7 protein expression and TLR7-driven production of IFN-α by pDCs in females but not in males (Azar et al, 2020). Our aim was to investigate whether this SNP was also functional in pDCs from HIV-1–infected female under ART with sustained undetectable viral load and to compare the TLR7-driven

cytokine production of pDCs between uninfected controls and HIV/ART females homozygous for the rs179008 major c.32A allele. HIV/ART women with a known A/A or A/T genotype from the ANRS EP53 X_LIBRIS cohort were reassessed for the relative frequency of circulating pDCs and the functional response of their blood pDCs following stimulation with TLR7 ligands using the flow cytometry strategy described in Fig S1 as previously described (Azar et al, 2020). We observed a lower frequency of pDCs in the blood of HIV/ART women relative to age-matched uninfected controls (UCs) (Fig 1A), as per previous reports (Kamga et al, 2005; Fontaine et al, 2009; Sabado et al, 2010). Among HIV/ART and UC women, pDC frequencies in the AA and AT subjects were equivalent (Fig 1B). Fresh PBMCs were stimulated with DOTAP vesicles loaded with Gag$_{RNA1166}$, a synthetic HIV-1–derived RNA ligand of TLR7 and TLR8. In agreement with our previous work with healthy female donors, the frequency of IFN-α–producing pDCs in Gag$_{RNA1166}$-stimulated PBMCs was significantly reduced in AT heterozygous patients compared with AA HIV/ART women (Fig 1C). Thus, the rs179008 c.32A>T pQTL is also a functional polymorphism controlling TLR7-driven production of IFN-I by pDCs in HIV-1/ART women.

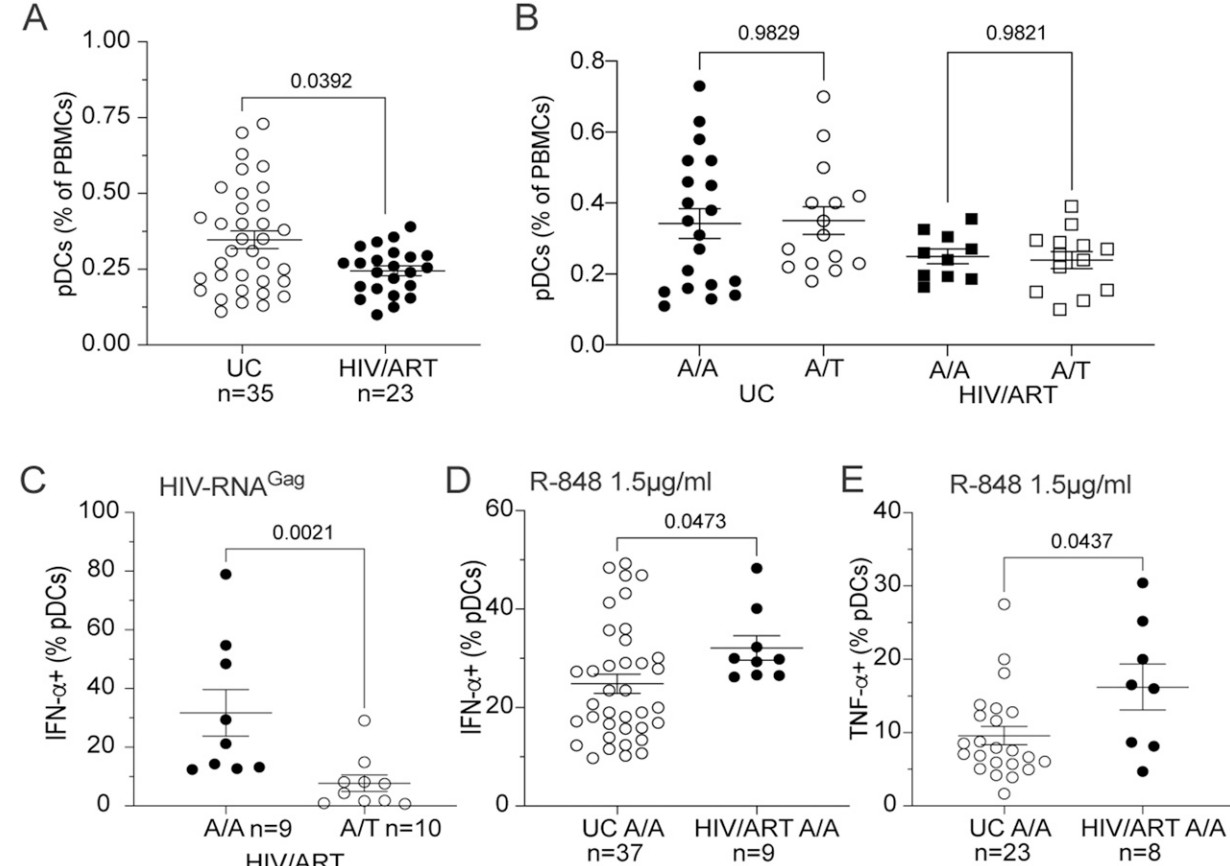

**Figure 1. pDCs from HIV/ART women exhibited increased functional responses to TLR7 ligands compared with UCs.**
**(A)** PBMCs from HIV/ART females and age-matched uninfected controls (UCs) were monitored for the percentage of pDCs (lin⁻ CD123⁺BDCA4⁺) as indicated in Fig S1.
**(B)** The frequency of blood pDCs are represented according to the expression of the indicated genotype of the rs179008 SNP (A/A versus A/T) of *TLR7*. **(C)** Fresh PBMCs from the HIV/ART women with the indicated rs179008 genotypes were stimulated with HIV-derived Gag$_{RNA1166}$ in DOTAP and intracellularly stained for IFN-α expression in CD123⁺BDCA4⁺ pDCs. **(D, E)** Fresh PBMCs from HIV/ART and UC females homozygous for the *TLR7* rs179008 A/A SNP were stimulated with TLR7/8 ligand R-848 (1.5 μg/ml) and intracellularly stained for IFN-α and TNF-α. **(D, E)** Frequencies of pDCs producing IFN-α (D) or TNF-α (E) are shown. **(A, B, C, D, E)** Errors bars represent the mean ± SEM. Data from individual subjects are shown. Statistical analysis was performed using the Mann–Whitney test. *P*-values are shown. Source data are available for this figure.

## Chronic HIV-1 infection is associated with the enhanced functional state of TLR7-stimulated pDCs in HIV/ART women carrying the rs179008 AA SNPs

As TLR7-driven functional response of pDCs has been reported to remain intact under ART (Chang et al, 2012; Tsai et al, 2017), we next compared the frequency of IFN-*α*– and TNF-*α*–producing pDCs in response to optimal stimulation with the TLR7/8 ligand Resiquimod (R-848) in HIV/ART women compared with age-matched UCs, both homozygous for the frequent rs179008 A allele. As shown in Fig 1D and E, pDCs from HIV/ART females or UCs were efficiently stimulated to produce IFN-*α* and TNF-*α*. Of note, the average frequency of IFN-$\alpha^+$ pDCs was significantly higher in HIV/ART females compared with uninfected controls (Fig 1D). The same trend was found for the TNF-*α* response (Fig 1E), suggesting that the NF-kB pathway leading to pro-inflammatory cytokine production downstream of TLR7 was also substantially increased in pDCs from HIV/ART women.

## X chromosome inactivation escape of TLR7 in pDCs is similar between HIV/ART and UC women

*TLR7* is encoded on the X chromosome and has been shown to escape XCI in a substantial proportion of immune cells from healthy women (Souyris et al, 2018; Hagen et al, 2020). However, whether the status of XCI escape can be affected by chronic HIV-1 infection has never been examined. Because the escape from XCI of the *TLR7* gene was associated with enhanced functional characteristics in female B cells (Souyris et al, 2018) and pDCs (Hagen et al, 2020), we decided to design a single-cell RT–qPCR approach to measure the frequency of *TLR7* biallelic cells among pDCs from HIV/ART women. We tested the hypothesis that the enhanced TLR7-driven responsiveness of pDCs in this population could be associated with a higher frequency of cells with escape from XCI of *TLR7* gene compared with uninfected subjects. The experimental workflow is depicted in Fig S2. To tag the expression of *TLR7* gene, we used heterozygous female donors expressing the rs179008 A/T and the rs3853839 G/C SNPs (Figs 2A and S2) as previously reported (Souyris et al, 2018). Frozen PBMCs were cultured overnight in the presence of IFN-*β* as we found that this was associated with enhanced pDC recovery compared with the condition without cytokine (Fig S3). Moreover, type 1 IFN signaling has been suggested to promote optimal expression of *TLR7* at single-cell resolution (Hagen et al, 2020). pDCs were single-cell sorted into two 96-well plates to have at least between 100-200 pDCs analyzed in this assay using the gating strategy shown in Fig S3. Indeed, we found that the cumulative count of single-cell sorted pDCs required to achieve robust determination of allelic expression of *TLR7* was already achieved at cell count ≥ 100 cells (Fig S2C and D). By using the ratio of allele-specific fluorescence signals in the KASP PCR (Fig S2B), we were able to measure the relative abundance of *TLR7* transcripts derived from either allele on each chromosome at a single-cell level. Each cell was then classified into mono- or biallelic cell for *TLR7* expression when the minor allele frequency was estimated to range below or above 10% of the relative proportion of *TLR7* transcripts, respectively (Fig S2B). The allelic expression of *TLR7* using the simultaneous determination of both SNPs from each single-cell from two double-heterozygous female donors bearing distinct haplotypes

are shown in Fig 2B–D. The allele-of-origin profiles were plotted according to the expression of the rs3853839 3' UTR SNP into mono-allelic C or G cells and biallelic GC cells (Fig 2C–E). Within each subset, we then reported the results regarding allelic expression of the second rs179008 A/T SNPs to determine the haplotypic association of both SNPs. For the female subject bearing the A-C/T-G haplotype (Fig 2B and C), most of the cells with mono-allelic expression of C or G allele of rs3853839 were also positive for either the A or the T allele of rs179008, respectively, indicating that the A-C and the T-G SNPs were located on the same X chromosome (Fig 2C). However, some single cells (<5%) were also positive for the SNPs belonging to the alternative X chromosome and were reassigned as biallelic cells. Similarly for the second female subject, using the same strategy, we established that this female donor was bearing the T-C/A-G haplotype (Fig 2D and E). Haplotype inference and allelic reassignment resulting in the final determination of haplotype profiling for all donors are shown in Fig 2F and G. Using this approach, we found a significant underestimation of biallelic cell frequency when each single SNP were independently used for allele of origin measurement compared with the joined analysis of the diplotype SNPs (Fig 2G). By contrast, no significant differences in mono-allelic cell frequencies were observed using each SNP individually or as haplotype (Fig 2F). Together, these results demonstrate that the parallel analyses of two SNPs from the same double-heterozygous female donors give an accurate and concordant estimation of the frequency of cells with mono- and biallelic expression of *TLR7*.

We then compared the frequency of cells with biallelic expression of *TLR7* in single-cell sorted pDCs from female heterozygous for the rs179008 and rs3853839 SNPs (UC: n = 8; HIV/ART: n = 6). The frequencies of pDCs with biallelic expression of *TLR7* were ranging from 8 to 32% in the 14 females tested, and no significant differences were observed between UC and HIV/ART women (Fig 2G and H). We also analyzed the frequency of biallelic cells in females heterozygous for the rs3853839 G/C SNP. Again, using this single SNP to tag allelic expression, we also found similar frequencies of biallelic cells in pDCs from HIV-1–infected women compared with UC women (Fig 2H). Combined, these results represent the largest quantitative assessment of the frequency of *TLR7* biallelic cells in human immune cells with 29 subjects examined in total. In agreement with previous works (Souyris et al, 2018; Hagen et al, 2020), this demonstrates that escape from XCI of *TLR7* is a common and highly reproducible feature in female pDCs not only in UC but also in HIV/ART women. Differences in the frequency of pDCs with biallelic expression of *TLR7* are unlikely to explain the enhanced functional responses observed in HIV/ART women (Fig 1).

## Chronic HIV-1 infection is associated with higher expression levels of TLR7 mRNA at single-cell resolution

Because our experimental workflow allows us to quantify the mRNA expression level from each allele of *TLR7* (Fig S2A), we compared the expression levels of *TLR7* mRNA expressed from each X chromosome in mono- and biallelic cells in single-cell sorted pDCs. In mono-allelic cells, *TLR7* mRNA relative expressions were similar between the two X chromosomes in all the donors, regardless of the expression of the rs179008 or rs3853839 SNPs. This was observed

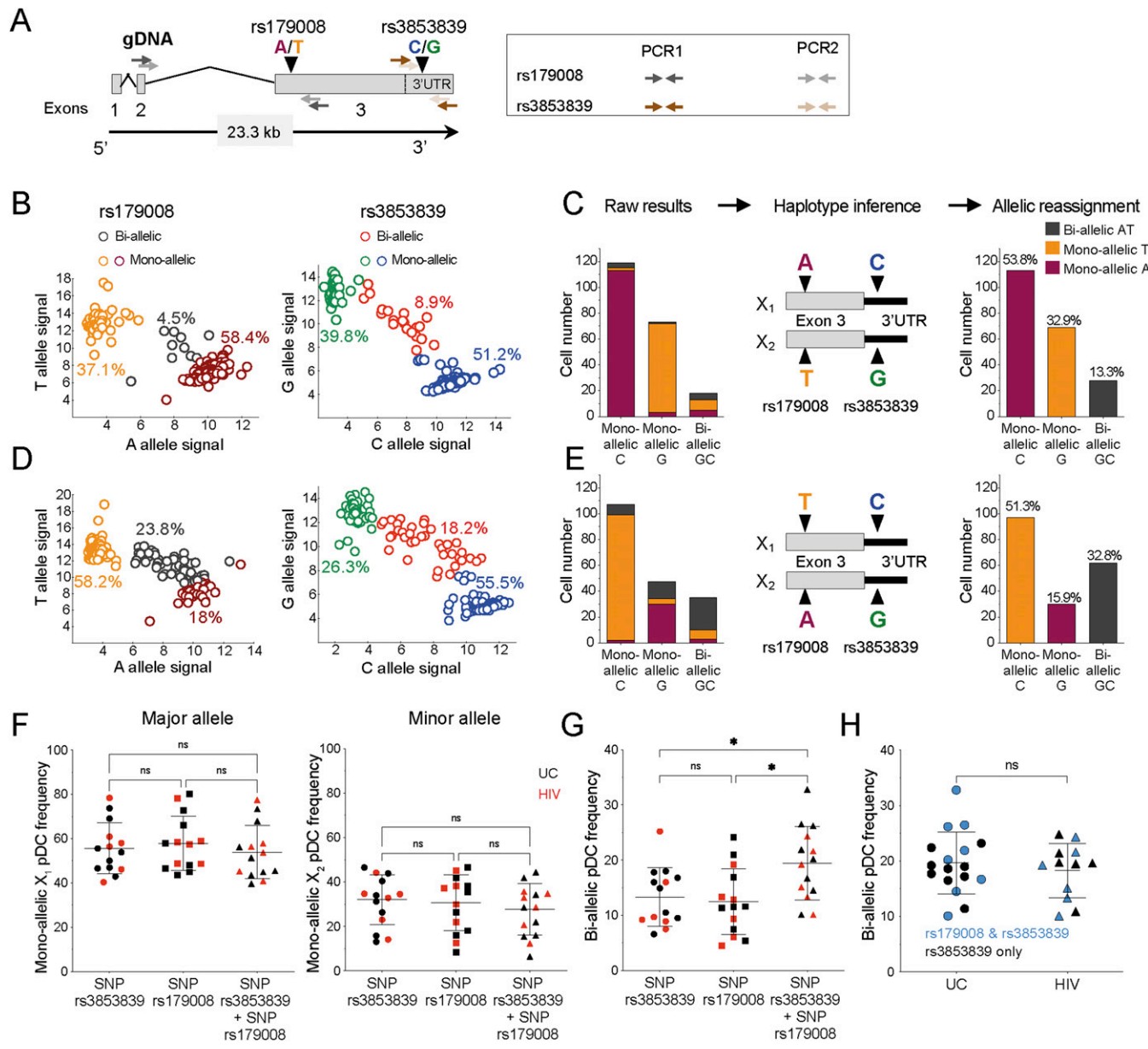

**Figure 2. The frequencies of pDCs with biallelic expression of *TLR7* are similar between HIV/ART and UC women.**
**(A)** Map of *TLR7* locus with SNPs rs179008 (A/T) and rs3853839 (C/G) and the positioning of the primers used for sequential PCR amplification of *TLR7* cDNA at single-cell resolution. **(B, C, D, E)** pDCs were single-cell sorted from two representative women heterozygous for both SNPs, displaying various proportion of biallelic cells. **(B, D)** Representative allele-of-origin profile of individual pDCs for the SNPs *TLR7* rs3853839 (right panel) and rs179008 (left panel) from two different heterozygous female donors. Each dot represents one cell with mono-allelic or biallelic expression of *TLR7*. **(B, C, D, E)** Raw results, haplotype inference, and cell reassignment for the proportion of pDCs mono-allelic A, T, or biallelic AT pDCs within mono-allelic C, G, or biallelic GC for *TLR7* from two representative heterozygous women, bearing either the A-C/T-G (B, C) or the T-C/A-G (D, E) haplotypes. **(F, G)** Frequencies of mono-allelic (F) or biallelic (G) pDCs from healthy (n = 8) and HIV/ART (n = 6) women by using the SNP rs3853839 and SNP rs179008 alone or in combination. Statistical analysis was performed using a one-way ANOVA followed by a Tukey's multiple comparisons test. ns, not significant. Exact *P*-values are shown. Errors bars represent the mean ± SEM. **(H)** Frequencies of *TLR7* biallelic pDCs from UCs or HIV/ART determined using the *TLR7* SNPs rs3853839 and rs179008 haplotypes or the rs3853839 SNP G/C only. The *t* test was used for statistical analysis. ns, not significant. Errors bars represent the mean ± SEM.
Source data are available for this figure.

not only in UCs but also in HIV/ART women, whatever the haplotype-borne (Fig S4A and B). These results demonstrate, at single-cell resolution, that carriage of one or the other of the minor allele rs179008 A/T or the rs3853839 G/C had no measurable impact on *TLR7* mRNA expression in female pDCs (Fig S4C and D). This was

expected for rs179008 T allele from our previous work (Azar et al, 2020), but much less for rs3853839 C allele which was previously reported to control the binding of the microRNA-3148 and *TLR7* mRNA degradation (Deng et al, 2013). Our results now show that miR-3148 binding to *TLR7* 3'-UTR segment bearing the C allele is

unlikely to control *TLR7* mRNA level in female pDCs. Our results also indicate that the expression of *TLR7* from the active X chromosome (Xa) is similar in the mosaic population of mono-allelic pDCs whatever the parent of origin of the Xa (Xm or Xp), contrary to what has been suggested in mice (Golden et al, 2019). We therefore decided to pool the data from the mono-allelic populations and compared them to the *TLR7* expression level detected in *TLR7* biallelic cells. Both in UCs and in HIV/ART women, we found significantly higher *TLR7* mRNA transcript levels in pDCs with biallelic than those with mono-allelic *TLR7* expression (Fig 3). Although, these differences were low (1.13- to 1.14-fold in UCs; 1.25–1.33-fold in HIV/ART), they were consistent with expression of this gene from both X chromosomes, including the Xi, in agreement with previous works in B cells (Souyris et al, 2018) and pDCs (Hagen et al, 2020).

Unexpectedly, we observed that the relative expression of *TLR7* at single-cell resolution was substantially higher in pDCs from HIV/ART females compared with UCs regardless of the SNP studied (Fig 3A and B). The average fold-change was calculated, and we estimated that *TLR7* gene expression from the Xa (mono-allelic cells) was 1.5-fold higher in average in pDCs from HIV/ART women. A similar trend was also observed in biallelic pDCs with higher

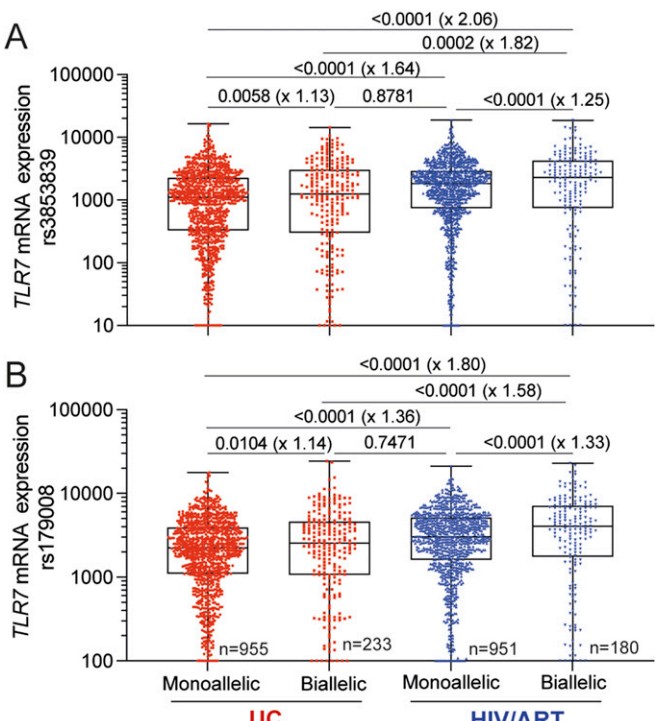

**Figure 3. Single-cell analysis reveals enhanced *TLR7* mRNA expression in pDCs from HIV-infected women under ART.**

**(A, B)** Relative expression of *TLR7* mRNA in pDCs from UCs or HIV/ART women with either mono-allelic or biallelic expression of *TLR7* were calculated at single-cell resolution using primers specific for either the rs3853839 (A) or the rs179008 (B) cDNA sequence of *TLR7* as shown in Figs S2 and 2. *TLR7* mRNA relative expression from pooled mono-allelic or biallelic pDCs from UCs (n = 8, n = 1,188, mean cell number/donor = 146) or HIV/ART (n = 6, n = 1,131; mean cell number/donor = 189) females. Statistical analysis was performed using a One-way ANOVA followed by Tukey's posttest corrected for multiple comparison. *P*-values and fold-change in median values in parenthesis are shown.
Source data are available for this figure.

expression levels of *TLR7* transcripts (1.58- to 1.82-fold higher) in biallelic cells from HIV/ART females compared with UCs (Fig 3A and B). We estimated that 18–20% of this increase was attributable to the additional *TLR7* mRNA expression from the Xi. Of note, the relative expression of *TLR7* mRNA in mono-allelic pDCs from HIV/ART females was similar to biallelic pDCs from UCs. Together, these results provide evidence for enhanced transcriptional activity of the *TLR7* locus from both the Xa and the Xi in HIV/ART women.

## Discussion

Together, our results show that although pDC number is lower in HIV/ART women, frequencies of IFN-α or TNF-α producing pDCs after TLR7 stimulation are higher than in healthy women. The analysis of *TLR7* XCI escape by single-cell RT–qPCR with SNPs markers demonstrated that the frequency of *TLR7* biallelic pDCs were similar between HIV/ART and UC women. *TLR7* biallelic pDCs expressed 1.13–1.25-time more *TLR7* mRNA than mono-allelic pDCs. This enhanced expression of *TLR7* mRNA transcripts in biallelic pDCs compared with mono-allelic ones was more pronounced in HIV/ART females than in UCs. Surprisingly, consistent with the increased innate cytokine responses of pDC from HIV/ART females, we observed a significant up-regulation of *TLR7* gene expression in pDCs including mono-allelic ones where *TLR7* was expressed almost exclusively from the Xa. These results provide evidence for an increased TLR7-driven pDC responsiveness associated with enhanced transcriptional activity of the *TLR7* locus on both X chromosomes, in pDCs from HIV/ART females.

It has been recently reported at single-cell resolution that female pDCs with escape from XCI in *TLR7* expressed not only higher levels of *TLR7* mRNA but also higher levels of transcripts coding for all IFN-α subtypes and IFN-β at steady state (Hagen et al, 2020). This observation suggested that the expression of the basal level of IFN-I mRNA could be a consequence of a higher *TLR7* mRNA expression, thereby discriminating functionally mono-allelic and biallelic pDCs (Hagen et al, 2020). Indeed, constitutive production of low levels of IFN-I in pDCs have been reported to act in an autocrine/paracrine manner to drive high levels production of IFN-I by pDCs (Kim et al, 2014). Thus, pDCs with high level expression of *TLR7* mRNA because of the cell-autonomous action of XCI escape of *TLR7* (Souyris et al, 2018; Hagen et al, 2020) could belong to a group of early responder pDCs with superior ability to produce IFN-I (Wimmers et al, 2018). This mechanism together with the enhanced transcriptional activity of the *TLR7* locus from the Xa observed in the general mono-allelic population in HIV/ART women could contribute to the functional reprogramming of pDCs in these subjects, by increasing the frequencies of pDCs with enhanced propensity to functionally respond to TLR7 agonist ligands. Of note, we found that the expression levels of *TLR7* mRNA in mono-allelic pDCs from HIV/ART females were similar to the levels measured in *TLR7* biallelic cells from UCs.

Our data corroborate recent observations where increased production of pro-inflammatory cytokines has been reported in monocytes stimulated through TLR4 or TLR7 in a large cohort of HIV-1-infected subjects under ART (van der Heijden et al, 2021). In this study, the authors observed a markedly increased monocyte-

derived cytokine response, mainly affecting IL-1β, upon stimulation with LPS, imiquimod, and *Mycobacterium tuberculosis* (van der Heijden et al, 2021). It was suggested that immune dysfunction and persistent inflammation in HIV/ART subjects because of microbial translocation or continuing HIV replication could promote functional adaptation of innate immune cells by epigenetic or metabolic reprogramming, a process known as "trained immunity" (Netea et al, 2020). Although epigenetic modifications have not been documented in HIV-infected subjects, future studies are warranted to investigate the mechanisms underlying the trained-immunity phenotype reported in monocytes (van der Heijden et al, 2021) and in pDCs (our present work).

The limitation of our study is that *TLR7* expression and pDC innate function was investigated in one sex. Whether our conclusion can be extrapolated to males will deserve dedicated studies. Of note, in the study by van der Heijden et al (2021), most of the HIV-1–infected subjects were males, and transcriptomic profiling of monocytes revealed broad up-regulation of inflammatory pathways, including enhanced *TLR7* mRNA (van der Heijden et al, 2021). Thus, the trained-immunity phenotype observed in male monocytes in this study could also possibly apply to male pDCs. However, it is important to note that sex differences exist regarding TLR7 protein expression and type I IFN production by pDCs, with reduced expression of both parameters in male cells compared with females (Meier et al, 2009; Seillet et al, 2012; Souyris et al, 2018; Azar et al, 2020). Whether this sex-bias in TLR7-driven pDC responsiveness could alter the trained-immunity–like phenotype we currently observed is still an open question. However, because the enhanced *TLR7* mRNA expression in pDCs from HIV/ART females was not affected by the mono-allelic expression of T allele of rs179008 T SNP, we can conclude that the functional expression of TLR7 protein is unlikely to contribute to the enhanced transcriptional activity of the *TLR7* locus and the trained-immunity–like phenotype we observed. We believe that epigenetic modifications at the *TLR7* locus in pDCs could be induced by pro-inflammatory cytokines, including IFN-I, which could be produced at low levels upon HIV-1 provirus reactivation whatever the sex (Kamada et al, 2018; Mitchell et al, 2020). In our study, analysis of TLR7 expression was performed after incubation with IFN-I; thus, we cannot exclude that changes in IFNAR-signaling between UC and HIV/ART subjects could also contribute to observed differences. Investigating the mechanisms underlying the functional reprogramming of pDCs will be warranted but highly challenging because of the scarcity of pDCs particularly in HIV-infected subjects. Besides myeloid cells, trained immunity have been reported in lymphoid cells, including NK cells, ILC1, ILC2, and DCs reviewed in Netea et al (2020). Our study is the first to report a functional reprogramming-like phenotype in human pDCs and could have important implication for the development of strategies targeting the HIV-1 latent reservoir by using TLR7 agonists in HIV/ART women.

In sum, we propose that in HIV/ART women, the innate function of pDCs is increased and is associated with markedly enhanced expression of the *TLR7* locus from both X chromosomes. Based on previous works by others showing strong association between *TLR7* mRNA expression and enhanced propensity of female pDCs to transcribe all IFN-I family members (Hagen et al, 2020), we believe that enhanced transcriptional regulation of *TLR7* mRNA could contribute to this trained-immunity–like phenotype through mechanisms that will deserve further investigation. Our data strengthen the interest of targeting the HIV-1 latent reservoir by using TLR7 agonists in HIV-1–infected women under ART and suggest that it will be critical to control for the expression of the rs179008 as this SNP could be associated with blunted pDC innate functions that still persist in HIV/ART women.

## Materials and Methods

### Donors and ethical compliance

Our study is in agreement with applicable French regulations and with the ethical principles of the Declaration of Helsinki. PBMCs of healthy blood from anonymous donors at the Toulouse blood transfusion center (Etablissement Français du Sang) were from a biobank authorized under agreement number 2-15-36 by the competent ethics board Comité de Protection des Personnes Sud-Ouest et Outre-Mer II, Toulouse. The cohort ANRS EP_53 X-LIBRIS (coordinator: Pr. P. Delobel, CHU Purpan) allows the study of HIV-1 donors has been previously described (Azar et al, 2020). The X-LIBRIS cohort was previously registered with ClinicalTrials.gov under identifier NCT01952587.

### Cell culture

Frozen PBMCs were defrosted and washed twice in R10 media which is complete RPMI 1640 medium supplemented with 2 mM L-glutamine, 1 mM sodium pyruvate, 100 U/ml penicillin/streptomycin, nonessential amino-acids, 50 μM 2-mercaptoethanol (all from Invitrogen) to which 10% heat-inactivated FBS (Sigma-Aldrich) was added. Cells were cultured overnight 37°C in 5% $CO_2$ air incubator. For single-cell RT–qPCR-KASP, after thawing, cells were cultured overnight with IFN-β 1 ng/ml in R10 complete medium.

### Cell stimulation analysis of cytokines production

Fresh PBMCs ($2.5 \times 10^6$ cells) were seeded in 24-well plate and stimulated either with 30 μg/ml HIV-1-derived Gag$_{RNA1166}$ synthetic oligoribonucletides (Eurogentec) complexed with DOTAP (Roche) or 1.5 μg/ml R-848 (Invivogen) for 5 h. Brefeldin A (eBiosciences) was added for the three last hours of culture. PBMCs were surface-labeled with anti-Lin-FITC (lin1; BD Biosciences), anti-BDCA4-APC (clone REA774; Miltenyi Biotec), and anti-CD123-PECy5 (clone 9R5; BD Biosciences). Cells were fixed in 2% paraformaldehyde and permeabilized in 0.5% saponin. Intracellular staining was performed using anti-IFNα-PE (clone LT27:295; Miltenyi Biotech) and anti-TNFα-AF700 (clone MAb11; BD Biosciences). Flow cytometry analysis was performed on LSRII instrument (BD Biosciences). Data were analyzed using the FlowJo software V10 (Tree Star).

### Flow cytometry cell sorting

Cells were incubated with anti-human CD32 (Fc gamma RII) (STEMCELL) for 5 min at 4°C. For cell surface markers, cells were

stained in MACS buffer (PBS with 1% FBS, 2 mM EDTA) for 30 min at 4°C with the following antibodies: CD14-VioBlue (clone REA599; Miltenyi Biotec), CD19-PEVio615 (clone LT19; Miltenyi Biotec), BDCA4-APC (clone REA774; Miltenyi Biotec), and CD123-PEVio770 (clone AC145; Miltenyi Biotec). Then cells were stained for viability with DRAQ7 (Abcam) for 5 min at 4°C. pDCs were sorted with a FACSAria II or FACS Aria-Fusion Cell Sorter (BD Biosciences) at one cell by well. Data were analyzed with the FlowJo software V10 (Tree Star).

### Genotyping and single-cell analysis of *TLR7* allelic expression

The workflow for single-cell cDNA analysis previously described in Souyris et al (2018) has been optimized further and is summarized in Fig S2. Briefly, pDCs from women heterozygous for the *TLR7* SNP rs3853839 and rs179008 were single-cell–sorted with a FACS Aria-Fusion Cell Sorter (BD Biosciences) in a 96-well plate preloaded with a medium containing 2% Triton X-100, 1 U/$\mu$l RNaseOut recombinant ribonuclease inhibitor (Thermo Fisher Scientific), 940 $\mu$M dNTPs, and 12.5 ng/$\mu$l random hexamer primers (Thermo Fisher Scientific). After cell sorting, single-cell lysates were subjected to RNA reverse transcription using 6.25 U/well Maxima H Minus reverse transcriptase (Thermo Fisher Scientific). Before KASP genotyping, both target SNPs were PCR-amplified using *TLR7* cDNA-specific primer pairs (Table S1), and negative wells screened out by real-time quantitative PCR (RT–qPCR) with nested primers (Table S1) and SsoAdvanced Universal SYBR Green Supermix (Bio-Rad Laboratories). The qPCR and KASP fluorescence-based assays were performed using the automated pipetting system epMotion 5070 (Eppendorf) and a Light Cycler 480 instrument (Roche). Relative allelic expression was calculated from the ratio of the FAM (allele C or A) and HEX (allele G or T) fluorescence signals from the respective KASP probes; a four-parameter standard curve was generated regularly for each Light Cycler 480 unit, using an R script based on package *drc* as described (Souyris et al, 2018). Standard curves were generated by mixing DNA samples from men of rs3853839 G/0 and C/0 genotypes or rs179008 A/0 and T/0 genotypes. Biallelic *TLR7* expression in a cell was inferred when the relative expression values of *TLR7* transcripts bearing the rs3853839 G allele or rs179008 T allele was comprised between 10% and 90%. A limit of detection (LOD) was defined as a Ct value of 23, and then all Ct values higher than the LOD were removed from the analysis. mRNA expression levels were defined as $2^{(LOD-Ct)}$ as described (Hagen et al, 2020).

### Statistical analysis

Statistical analyses and data presentation were performed using GraphPad Prism 9 software (La Jolla). Statistically significant differences between two groups were determined using two-tailed.

Mann–Whitney or unpaired *t* test was performed when indicated. Multiple comparisons were performed using the one-way ANOVA test followed by Tukey's posttest corrected for multiple comparisons as indicated. Values are reported as individual values and plotted as mean ± SEM or as median and IQ range. All statistical tests were two-tailed, and *P*-values <0.05 were considered to be statistically significant.

## Supplementary Information

## Acknowledgements

We gratefully acknowledge support from F. L'Faqihi, L De La Fuente, AL Iscache, V Duplan, and Hugo Garnier at the flow cytometry facility (INSERM U1291, INFINITY); PE Paulet at the Immunomonitoring core facility (INSERM U1291, INFINITY). The technical assistance of M Requena, M Cazabat, and R Carcénac (Toulouse University Hospital, INSERM U1291, INFINITY) is also greatly acknowledged. This work was supported by grants from the French National Agency for Research on AIDS and Viral Hepatitis (ANRS, EP-53 study) and SIDACTION (Grant 2018-1-AEQ-12035). F Abbas, P Azar, and A Youness were supported by fellowships from SIDACTION. A Youness was also supported by PhD fellowships from the CSL Behring Research Funds, Fondation des Treilles, and from the "Association de la Charité des jeunes de Kafarsir" (Libanon).

### Author Contributions

F Abbas: data curation, formal analysis, investigation, methodology, and writing—original draft, review, and editing.
C Cenac: data curation, investigation, methodology, and writing—original draft, review, and editing.
A Youness: data curation, formal analysis, and methodology.
P Azar: data curation, formal analysis, investigation, and methodology.
P Delobel: resources, funding acquisition, validation, and project administration.
J-C Guéry: conceptualization, data curation, formal analysis, supervision, funding acquisition, methodology, project administration, and writing—original draft, review, and editing.

### Conflict of Interest Statement

The authors declare that they have no conflict of interest.

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
