## [Reviewer comments · Life Science Alliance]

Life Science Alliance

HIV-1 infection enhances innate function and TLR7 expression in female plasmacytoid dendritic cells

Flora Abbas, Claire Cénac, Ali Youness, Pascal Azar, Pierre Delobel, and Jean-Charles Guéry

DOI: <https://doi.org/10.26508/lsa.202201452>

Corresponding author(s): Jean-Charles Guéry, INSERM U1291

Review Timeline:

Submission Date:	2022-03-17
Editorial Decision:	2022-04-26
Revision Received:	2022-07-22
Editorial Decision:	2022-08-12
Revision Received:	2022-08-19
Accepted:	2022-08-19

Scientific Editor: Novella Guidi

Transaction Report:

April 26, 2022

Re: Life Science Alliance manuscript #LSA-2022-01452-T

Dr. Jean-Charles Guery
INSERM U1291
Institut Toulousain des Maladies Infectieuses et Inflammatoires (INFINITY)
CHU Purpan
BP3028
Toulouse 31024
France

Dear Dr. Guery,

Thank you for submitting your manuscript entitled "HIV-1 infection induces functional reprogramming of female plasmacytoid dendritic cells associated with enhanced TLR7 expression" to Life Science Alliance. The manuscript was assessed by expert reviewers, whose comments are appended to this letter. We invite you to submit a revised manuscript addressing the Reviewer comments.

Thank you for this interesting contribution to Life Science Alliance. We are looking forward to receiving your revised manuscript.

Sincerely,

B. MANUSCRIPT ORGANIZATION AND FORMATTING:

Reviewer #1 (Comments to the Authors (Required)):

Abbas et al present an interesting study on pDC TLR7 expression in women infected with HIV and receiving ART. The group has previously reported very interesting biallelic expression of TLR7, due to incomplete X chromosome inactivation in some immune cells. This study follows on that previous work and examines this phenotype in the context of HIV infection under ART. TLR7 is a target for HIV viral clearance strategies, so the study is highly relevant and of interest to the field. However I have some questions, and concerns on some of their interpretation of their results.

1. Why were the two way statistical tests performed with non-parametric tests (Mann Whitney) while multi-group comparisons were performed with parametric tests (ANOVA) ? Similarly showing the median values (rather than means) is more appropriate for non-parametric data.
2. Why is there a difference in the groups numbers between Fig 1d & e ? are the IFN α + cells also TNF α + cells ?
3. Page 5, showing the response of the two cytokines does not support the following broad statement "that all pathways downstream of TLR7- signaling were substantially increased in HIV/ART women" which should be modified.
4. What was the rationale for culturing pDCs with IFN β , as opposed to IFN α for example, which they secrete?
5. Can the authors show representative allele-of-origin profile of individual pDCs from homozygous women? It would help to understand the distribution and how the cut offs are decided which is not very clear. Also, if the authors state « that the cumulative count of single-cell sorted pDCs required to achieve robust determination of allelic expression of TLR7 was already achieved at cell count {greater than or equal to} 100 cells » how is it possible to have single cell resolution? I'm clearly missing some information.
6. I don't think the authors show sufficient evidence to support the title that states "HIV-1 infection induces functional reprogramming", they compared uninfected with infected women at a single time point which does not allow to assign causality of this phenotype to the HIV infection, this title should be modified to better reflect their actual findings.

Minor

Grammatical errors

"has been found to be associated with enhanced functional characteristics in B cells from women (Souyris et al., 2018) and pDCs (Hagen et al., 2020)"

Reviewer #2 (Comments to the Authors (Required)):

The manuscript of Abbas et al. investigates TLR7 expression on plasmacytoid dendritic cells in HIV-1 infected/ART treated women and in healthy controls. The authors conclude that although pDCs are less in patients than in controls, their cytokine production is increased potentially due to an increase of TLR7 expression on patients' pDCs. TLR7 gene expression was increased in patients, whereas TLR7 silencing (or the lack of it) due to X chromosome inactivation was similar in patients and controls. The results are important for understanding pDC biology in the context of chronic HIV-1 infection but also due to a possibility of latent HIV reactivation by targeting TLR7.

In general, the manuscript is well presented and the results are clear and convincing. I would like to suggest a few points to consider/discuss further:

- 1) It wasn't entirely clear how the Results' first paragraph was connected to the rest of the data. The functional relevance for RS179008 polymorphism is demonstrated by the authors but it is not clear why this data is presented here, before the TLR7 expression analyses. The purpose of analysing the RS179008 polymorphism in TLR7-induced IFN production should be explained and the results more discussed.
- 2) Only a subset of pDCs produced IFN α or TNF in response to TLR7 stimulation, whereas the majority of the cells remained

negative.

One alternative explanation for the increased IFN α production in the HIV-1 infected cohort could be that pDCs are heterogeneous and the subset that does not produce cytokines in response to TLR7 stimulation might be selectively lost in the HIV-1 infected women. Then, from a lower number of total pDCs more would produce cytokines - irrespectively of their TLR7 level. Do the authors have data that excludes this scenario? Do the authors have data about the relationship of TLR7 expression/cells and their capacity to produce IFN α in response to TLR7 agonists? What are the cytokine non-producer pDCs? I feel that the heterogeneity of the pDCs in cytokine expression is a weak point of the work - the majority of the cells just doesn't seem to react to TLR7 stimulation, at least with the selected readout, and therefore the consequences of more or less TLR7 expression/cell are hard to evaluate.

3) The authors treat the thawed cells with type I IFN before evaluating TLR7 expression. Do the authors really study TLR7 expression differences in HIV-1 infected versus control women (at the steady-state) or do they study differences in IFN-induced TLR7 upregulation? This should be discussed as the interpretation of the results and the underlying mechanisms could be different.

Reviewer #3 (Comments to the Authors (Required)):

The report by Abbas and al show that pDCs from HIV/ART women have enhanced INF α and TNF α production following TLR7/TLR8 stimulation in comparison to healthy controls. This increase in cytokine response is associated with enhanced TLR7 transcription. Overall the paper is well written and the data are very clear.

Here are minor comments:

- What is the production of INF α and TNF α in pDCs from HIV+ females?
- Why using R848 (a TLR7 and TLR8 agonist) and not just a TLR7 agonist? Will the result be different?
- In HIV/ART women TLR7 mRNA expression is increased at the steady state, does it change upon R848 stimulation (5h)?
- Would it be possible to investigate TLR7 expression and trafficking in pDCs from HIV/ART females by performing some confocal microscopy?
- Fig EV1: B and C left panel, I was not aware that TLR7/8 stimulation induces upregulation of BDCA4 and CD123 on pDCs?

Point-by -point response to reviewer's comments

Reviewer #1 (Comments to the Authors (Required)):

Abbas et al present an interesting study on pDC TLR7 expression in women infected with HIV and receiving ART. The group has previously reported very interesting biallelic expression of TLR7, due to incomplete X chromosome inactivation in some immune cells. This study follows on that previous work and examines this phenotype in the context of HIV infection under ART. TLR7 is a target for HIV viral clearance strategies, so the study is highly relevant and of interest to the field. However, I have some questions, and concerns on some of their interpretation of their results.

1. Why were the two way statistical tests performed with non-parametric tests (Mann Whitney) while multi-group comparisons were performed with parametric tests (ANOVA) ? Similarly showing the median values (rather than means) is more appropriate for non-parametric data.

For the figure 2 F-G-H, we performed normality and lognormality tests (Anderson-Darling, D'Agostino & Pearson, Shapiro-Wilk and Kolmogorov-Smirnov) that all passed normality test, so we used an ordinary one-way ANOVA (F-G) and an unpaired t-test (Student) (H). For the figure 2H, we also tested with a nonparametric test (Mann-Whitney) and the result was also not significant.

2. Why is there a difference in the groups numbers between Fig 1d & e ? are the IFN α + cells also TNF α + cells ?

This was due to a technical issue as anti-TNF- α antibody was not included at the beginning of the study, where we focused on IFN- α . Thus, not all subjects were tested at the beginning for both cytokines. This is why the numbers are different between Fig 1D & E.

3. Page 5, showing the response of the two cytokines does not support the following broad statement "that all pathways downstream of TLR7- signaling were substantially increased in HIV/ART women" which should be modified.

In pDC TLR7 activates NF- κ B and IRF7 via Myd88 to induce inflammatory cytokines (TNF- α) and type I IFN (IFN- α), respectively. Both pathways were upregulated, but we agree with the reviewer and we have modified the sentence to make a more focus statement (see below).

"The same trend was found for the TNF- α response (Fig 1E) suggesting that the NF- κ B signaling pathways downstream of TLR7 were also substantially increased in HIV/ART women."

4. What was the rationale for culturing pDCs with IFN β , as opposed to IFN α for example, which they secrete?

IFN- β was found to improve pDCs survival after PBMC thawing and overnight culture (see Fig EV3). We have used this overnight culture system to synchronize and also optimize TLR7 expression in human PBMCs, particularly B cells in a previous study (Souyris M, *Sci Immunol* 2018; 3:eaap8855). We have not tested IFN- α in our assay, but it should work the same way as IFN- β .

5. Can the authors show representative allele-of-origin profile of individual pDCs from homozygous women? It would help to understand the distribution and how the cut offs are decided which is not very clear. Also, if the authors state « that the cumulative count of single-cell sorted pDCs required to achieve robust determination of allelic expression of TLR7 was already achieved at cell count {greater than or equal to} 100 cells » how is it possible to have single cell resolution? im clearly missing some information.

Individual profiles of hemizygous males were shown in our previous publication to show the cut-off and specificity of the assay (Figure 1 I; Souyris M, *Sci Immunol* 2018; 3:eaap8855). We calculated the relative proportions of the two alleles from the ratio of the respective endpoint fluorescence signals in the KASP PCR amplification with a cut-off fixed at 10% of the relative proportion of the minor allele (See Figure 1 C&D, Souyris M, *Sci Immunol* 2018; 3:eaap8855). Standard curves were generated by mixing DNA samples from men of rs3853839 G/0 and C/0 genotypes, or rs179008 A/0 and T/0 genotypes, as shown in Fig S2B.

We added these informations in the Material & Methods section (lane 375-377) and we modified the supplementary Fig S2B to show the cut offs at 90% and 10% for bi-allelic cell assignment.

The “single-cell RT-qPCR KASP” allow us to quantify for one single cell the relative expression of *TLR7* and its TLR7 allele of origin using the tag SNPs. If we combine all the single cells tested for one donor (>150 individual cells in average from two single-cell sorted 96-w plates), we can evaluate the frequency of bi-allelic cells among all the pDCs tested. Fig S2C shows the cumulative results of sc-RT-PCR-KASP assays from five 96-w plates containing one pDC/well from the same female donor. An average of 84 cells were analyzed per plates. As shown in Fig S2D, we found that for 100-200 cells tested, we can already achieve a reliable estimation of the frequencies of mono- and bi-allelic cells for a given subject.

6. I don't think the authors show sufficient evidence to support the title that states "HIV-1 infection induces functional reprogramming", they compared uninfected with infected women at a single time point which does not allow to assign causality of this phenotype to the HIV infection, this title should be modified to better reflect their actual findings.

We agree with the reviewer and we have changed the title. We hope the new title better reflects our findings.

Minor Grammatical errors

"has been found to be associated with enhanced functional characteristics in B cells from women (Souyris et al., 2018) and pDCs (Hagen et al., 2020)" Page 5

This has been corrected.

Reviewer #2 (Comments to the Authors (Required)):

The manuscript of Abbas et al. investigates TLR7 expression on plasmacytoid dendritic cells in HIV-1 infected/ART treated women and in healthy controls. The authors conclude that although pDCs are less in patients than in controls, their cytokine production is increased potentially due to an increase of TLR7 expression on patients' pDCs. TLR7 gene expression was increased in patients, whereas TLR7 silencing (or the lack of it) due to X chromosome inactivation was similar in patients and controls. The results are important for understanding pDC biology in the context of chronic HIV-1 infection but also due to a possibility of latent HIV reactivation by targeting TLR7.

In general, the manuscript is well presented and the results are clear and convincing. I would like to suggest a few points to consider/discuss further:

1) It wasn't entirely clear how the Results' first paragraph was connected to the rest of the data. The functional relevance for RS179008 polymorphism is demonstrated by the authors but it is not clear why this data is presented here, before the TLR7 expression analyses. The purpose of analysing the RS179008 polymorphism in TLR7-induced IFN production should be explained and the results more discussed.

We have previously reported that the rs179008 T allele is a functional polymorphism reducing TLR7 protein expression and the TLR7-driven IFN α response of female pDCs even in heterozygous female pDCs (Azar, JCI Insight 2020). This SNP had no impact on TLR7 mRNA expression. This is a sex-specific pQTL (no effect in males) which represents a confounding factor when studying the functional response of female pDCs. The aim of the results first paragraph was 1) to investigate whether this SNP was also functional in pDCs from HIV-1-infected female under ART, and 2) to compare the TLR7-driven cytokine production of pDCs between uninfected females and HIV/ART females.

We have clarified this in the first paragraph in our revised manuscript.

2) Only a subset of PDCs produced IFN α or TNF in response to TLR7 stimulation, whereas the majority of the cells remained negative.

One alternative explanation for the increased IFN α production in the HIV-1 infected cohort could be that PDCs are heterologous and the subset that does not produce cytokines in response to TLR7 stimulation might be selectively lost in the HIV-1 infected women. Then, from a lower number of total PDCs more would produce cytokines - irrespectively of their TLR7 level. Do the authors have data that excludes this scenario? Do the authors have data about the relationship of TLR7 expression/cells and their capacity to produce IFN α in response to TLR7 agonists? What are the cytokine non-producer PDCs? I feel that the heterogeneity of the PDCs in cytokine expression is a weak point of the work - the majority of the cells just doesn't seem to react to TLR7 stimulation, at least with the selected readout, and therefore the consequences of more or less TLR7 expression/cell are hard to evaluate.

It is not surprising that only a fraction of pDCs produce IFN- α in this type of assay. The reason for that is not completely understood. As mentioned in the discussion (p 9 lane 258-259), analysis of early pDC responses at single cell resolution have shown that a small population of pDCs are able to respond early to TLR stimulation to produce IFN- α (Wimmers et al, 2018 Nat Comm 9:3317). It has been proposed that stochastic gene regulation could be responsible for

type I IFN but not TNF α . At the population level, it was shown that the vigorous response can be driven by a type I IFN-dependent amplification loop (Wimmers et al, 2018 Nat Comm 9:3317). However, in this study the authors examined the response to TLR9 ligands, which is less effective than R848 to induce IFN- α production in such short-term stimulation assay.

In our hand, the frequencies of IFN- α producing pDCs ranged from 10 to 50% of cells, which is in agreement with previous works by us (Seillet et al., Blood 2012; 119:454-64; Azar et al., JCI Insight 2020; 5:e136047) and others (Meier et al., Nat. Med. 2009; 15:955-9) using TLR7-ligands. This is not so rare and as shown now in the supplementary Fig S1, we noticed that TLR7 activation induced up-regulation of CD123 and BDCA-4 in a homogenous manner suggesting that all pDCs are actually sensitized by R848, but only some of them can produce IFN- α (see Figure for reviewer #3, below). As shown in this Figure, CD123 and BDCA-4 were homogeneously upregulated (as HLA-DR molecules, not shown) in all pDCs activated through TLR7 with R848 or R837. This shows that all pDCs can sense the TLR7-ligands, but only some of them produce the cytokines.

3) The authors treat the thawed cells with type I IFN before evaluating TLR7 expression. Do the authors really study TLR7 expression differences in HIV-1 infected versus control women (at the steady-state) or do they study differences in IFN-induced TLR7 upregulation? This should be discussed as the interpretation of the results and the underlying mechanisms could be different.

We agree with the reviewer comment. We cannot exclude this scenario. This point has been discussed page 10 lane 295-298.

Reviewer #3 (Comments to the Authors (Required)):

The report by Abbas and al show that pDCs from HIV/ART women have enhanced INF α and TNF α production following TLR7/TLR8 stimulation in comparison to healthy controls. This increase in cytokine response is associated with enhanced TLR7 transcription. Overall, the paper is well written and the data are very clear.

Here are minor comments:

- What is the production of INF α and TNF α in pDCs from HIV+ females?

In this study, the functional assay was performed on fresh PBMCs obtained from blood samples with a limited volume. It was not possible to obtain sufficient amount of blood to purify pDCs and then assess the production of cytokines. We focused on the intracellular staining as this requires fewer cells and can be done on whole PBMCs. Moreover, it allows the direct visualization of cytokines inside the cells and give an estimate of the frequency of responding cells.

- Why using R848 (a TLR7 and TLR8 agonist) and not just a TLR7 agonist? Will the result be different?

In a previous study, we have shown that R848 is a much more potent inducer of IFN- α as compared to R837 (Azar et al., JCI Insight 2020; 5:e136047). In fact, R838 induces preferentially IFN α + pDCs and low frequency of TNF α + pDCs, whereas it is the opposite for R837. Even if R848 activates both TLR7&TLR8, pDCs do not express TLR8 and we preferred to use R848 ligand to focus on the IFN α response.

- In HIV/ART women TLR7 mRNA expression is increased at the steady state, does it change upon R848 stimulation (5h)?

The expression of TLR7 at single-cell resolution was not tested upon R848 stimulation.

- Would it be possible to investigate TLR7 expression and trafficking in pDCs from HIV/ART females by performing some confocal microscopy?

We agree with the reviewer that it would be interesting to investigate the trafficking and subcellular localization of TLR7 in pDCs from HIV/ART women compare to uninfected women. Unfortunately, we are not aware of any TLR7-specific antibody suitable for confocal microscopy or FACS analysis of human TLR7.

- Fig EV1: B and C left panel, I was not aware that TLR7/8 stimulation induces upregulation of BDCA4 and CD123 on pDCs?

We measured the fluorescence intensity (GMFI) of BDCA4 and CD123 on different donors, upon several stimulation with various TLR7 ligands and we confirmed the upregulation of those two markers on pDCs upon 5-hour activation (see Figure below for the reviewer). However, we believe that this observation is beyond the scope of our study, and we decided not to include those results to the revised manuscript.

[Figure removed by LSA Editorial Staff per authors' request.]

Figure for reviewers. CD123 and BDCA-4 are homogenously upregulated in TLR7-activated pDCs. Freshly isolated PBMCs (n = 8) were stimulated for 5 hours with the indicated TLR7/8 ligand R-848 (1.5 µg/ml) or R-837 (1.5 µg/ml) or left untreated and processed as in Fig S1. The gating strategy for analysis of human pDCs is shown in Fig S1. pDCs were identified as Singlet, CD14^{neg}CD19^{neg}CD123⁺ BDCA-4⁺ cells. FSC/SSC (**A**) and BDCA4/CD123 (**B**) contour plots of resting and R848-activated pDCs. GMFI of BDCA-4 (**C**) or CD123 (**D**) in pDCs following activation with R-848 or R-837. One-way ANOVA followed by Tukey multiple comparison test. *, p < 0.05; **, p<0.01.

August 12, 2022

RE: Life Science Alliance Manuscript #LSA-2022-01452-TR

Dr. Jean-Charles Guéry
INSERM U1291
Toulouse Institute for Infectious and Inflammatory Diseases (INFINITY)
CHU Purpan
BP3028
Toulouse Cedex 3 31024
France

Dear Dr. Guéry,

Thank you for submitting your revised manuscript entitled "HIV-1 infection enhances innate function and TLR7 expression in female plasmacytoid dendritic cells". We would be happy to publish your paper in Life Science Alliance pending final revisions necessary to meet our formatting guidelines.

- please add the Twitter handle of your host institute/organization as well as your own or/and one of the authors in our system
- we encourage you to introduce your panels in your figure legends in alphabetical order
- please upload your table files in editable excel or doc file format
- please double-check your figure callouts for your supplementary figure1 and use the format: Fig S1 rather than Fig EV1

A. FINAL FILES:

B. MANUSCRIPT ORGANIZATION AND FORMATTING:

Sincerely,

Reviewer #1 (Comments to the Authors (Required)):

The authors have sufficiently addressed my concerns.

Reviewer #2 (Comments to the Authors (Required)):

The manuscript of Abbas et al. investigates TLR7 expression on plasmacytoid dendritic cells in HIV-1 infected/ART treated women and in healthy controls. The authors conclude that although pDCs are less in patients than in controls, their cytokine production is increased potentially due to an increase of TLR7 expression on patients' pDCs. TLR7 gene expression was increased in patients, whereas TLR7 silencing (or the lack of it) due to X chromosome inactivation was similar in patients and controls. The results are important for understanding pDC biology in the context of chronic HIV-1 infection but also due to a possibility of latent HIV reactivation by targeting TLR7. In general, the manuscript is well presented and the results are clear and convincing.

The authors have addressed my concerns and I have no further questions.

Reviewer #3 (Comments to the Authors (Required)):

The authors have answered my comments and the paper is now suitable for publication.

August 19, 2022

RE: Life Science Alliance Manuscript #LSA-2022-01452-TRR

Dr. Jean-Charles Guéry
INSERM U1291
Toulouse Institute for Infectious and Inflammatory Diseases (INFINITY)
CHU Purpan
BP3028
Toulouse Cedex 3 31024
France

Dear Dr. Guéry,

Thank you for submitting your Research Article entitled "HIV-1 infection enhances innate function and TLR7 expression in female plasmacytoid dendritic cells". It is a pleasure to let you know that your manuscript is now accepted for publication in Life Science Alliance. Congratulations on this interesting work.

DISTRIBUTION OF MATERIALS:

Again, congratulations on a very nice paper. I hope you found the review process to be constructive and are pleased with how the manuscript was handled editorially. We look forward to future exciting submissions from your lab.

Sincerely,
